# Learning epistatic gene interactions from perturbation screens

**Kieran Elmes** [1][ORCID][☯], **Fabian Schmich** [2,3][ORCID][☯], **Ewa Szczurek** [4][ORCID], **Jeremy Jenkins** [5][ORCID], **Niko Beerenwinkel** [2,3][ORCID]*, **Alex Gavryushkin** [1,6][ORCID]*

**1** Department of Computer Science, University of Otago, Dunedin, New Zealand, **2** Department of Biosystems Science and Engineering, ETH Zurich, Basel, Switzerland, **3** SIB Swiss Institute of Bioinformatics, Basel, Switzerland, **4** Institute of Informatics, University of Warsaw, Warsaw, Poland, **5** Novartis Institutes for BioMedical Research, Cambridge, Massachusetts, United States of America, **6** School of Mathematics and Statistics, University of Canterbury, Christchurch, New Zealand

☯ These authors contributed equally to this work.
* niko.beerenwinkel@bsse.ethz.ch (NB); alex@biods.org (AG)

**Data Availability Statement:** Everything necessary to reproduce our results, including data, has been uploaded to Github: https://github.com/bioDS/xyz-simulation.

**Funding:** This work has partially been funded by SystemsX.ch, the Swiss Initiative in Systems

## Abstract

The treatment of complex diseases often relies on combinatorial therapy, a strategy where drugs are used to target multiple genes simultaneously. Promising candidate genes for combinatorial perturbation often constitute epistatic genes, i.e., genes which contribute to a phenotype in a non-linear fashion. Experimental identification of the full landscape of genetic interactions by perturbing all gene combinations is prohibitive due to the exponential growth of testable hypotheses. Here we present a model for the inference of pairwise epistatic, including synthetic lethal, gene interactions from siRNA-based perturbation screens. The model exploits the combinatorial nature of siRNA-based screens resulting from the high numbers of sequence-dependent off-target effects, where each siRNA apart from its intended target knocks down hundreds of additional genes. We show that conditional and marginal epistasis can be estimated as interaction coefficients of regression models on perturbation data. We compare two methods, namely `glinternet` and `xyz`, for selecting non-zero effects in high dimensions as components of the model, and make recommendations for the appropriate use of each. For data simulated from real RNAi screening libraries, we show that `glinternet` successfully identifies epistatic gene pairs with high accuracy across a wide range of relevant parameters for the signal-to-noise ratio of observed phenotypes, the effect size of epistasis and the number of observations per double knockdown. `xyz` is also able to identify interactions from lower dimensional data sets (fewer genes), but is less accurate for many dimensions. Higher accuracy of `glinternet`, however, comes at the cost of longer running time compared to `xyz`. The general model is widely applicable and allows mining the wealth of publicly available RNAi screening data for the estimation of epistatic interactions between genes. As a proof of concept, we apply the model to search for interactions, and potential targets for treatment, among previously published sets of siRNA perturbation screens on various pathogens. The identified interactions include both known epistatic interactions as well as novel findings.

Biology, under IPhD grant 2009/025 and RTD grants 51RT-0 126008 (InfectX) and 51RTP0 151029 (TargetInfectX), evaluated by the Swiss National Science Foundation. AG and KE acknowledge support from the Royal Society Te Aparangi through a Rutherford Discovery Fellowship (RDF-UOO1702) awarded to AG. AG and KE were partially supported by Ministry of Business, Innovation, and Employment of New Zealand through an Endeavour Smart Ideas grant (UOOX1912) and a Data Science Programmes grant (UOAX1932).

**Competing interests:** JJ is an employee of Novartis Institutes for BioMedical Research, which provided support in the form of the salary, but did not have any additional role in the study design, data collection and analysis, decision to publish, or preparation of the manuscript. This does not alter our adherence to PLOS ONE policies on sharing data and materials.

# 1 Introduction

Genetic interactions are also referred to as epistasis, a term that originates from the field of statistical genetics and describes genetic contributions to the phenotype that are not linear in the effects of single genes [1, 2]. Considering two genes at a time, positive and negative epistasis refer to a greater and smaller effect, respectively, of the double mutant genotype than expected from the two single mutant genotypes relative to the wild type. In genetics, the phenotype of primary interest is the reproductive success of a cell, which is commonly termed fitness [3]. In this context, a fitness landscape is the mapping of each combination of possible configurations of gene mutations to a fitness phenotype [4].

The knowledge of fitness landscapes is highly relevant for personalized disease treatment [5]. In cancer, for example, genetic aberrations result in cells with increased somatic fitness, for instance, by evading apoptosis or gaining the ability to metastasise. This increase subsequently promotes post-metastatic tumour development [6]. A major challenge in cancer therapy is the fact that many genes with driving mutations cannot be adequately targeted for inhibition due to toxic side effects and rapid development of drug resistance [7, 8]. To overcome this challenge, a strategy based on the inhibition of genes that interact with genes with cancer driving alterations was proposed [9]. This strategy is based on the principle of synthetic lethality [5, 10, 11], the extreme case of negative epistasis, where single mutants are compatible with cell viability but the double mutant results in cell death. Identifying synthetic lethal gene interactions allows targeting cancer cells in which one of the two genes is mutated, by using drugs that affect the other. In the presence of this drug, the cancer cell lineage will no longer be viable [12].

The identification of fitness landscapes is however a very challenging task, simply due to the exponential growth of the space of interactions. For yeast, for example, it has been shown to be feasible to experimentally perform 75% of all pairwise knockouts [13]. Similarly, [14, 15] study fitness landscapes with a small number of genes in which all or nearly all genotypes of interest have been measured. However, in humans, with approximately 20,000 protein-coding genes, this would constitute to almost 200 million experiments to test all pairwise interactions. An approach that has been successfully applied to identify synthetic lethality *in vitro* is large-scale perturbation screening of human cancer cell lines using RNA interference [16–19]. However, this strategy only allows cataloguing synthetic lethal gene pairs where one gene is always specific to the screened cell line. While these methods may be sufficient for the identification of a few promising targets for cancer therapy, they do not allow us to estimate general pairwise gene interactions at the human exome scale. To our knowledge, there are currently no methods for inferring gene interactions at this scale. We therefore focus on demonstrating that our approach is sound, rather than comparing to existing methods.

Short-interfering RNAs (siRNAs), the reagents used in RNAi perturbation screening, exhibit strong off-target effects, which results in high numbers of false positives rendering the perturbations hard to interpret [20]. While this is usually conceived as a problem, here we take advantage of this property for the estimation of genetic interactions [21–23]. We propose a novel approach for the second order approximation of a human fitness landscape by inferring the fitness of single gene perturbations and their pairwise interactions from RNAi screening data (Fig 1). Our approach is not restricted to interactions with mutant genes of a specific cell line or explicit double knockdowns. We leverage the combinatorial nature of sequence-dependent off-target effects of siRNAs, where each siRNA in addition to the intended on-target knocks down hundreds of additional genes simultaneously. Not distinguishing between on- and off-targeted genes, we consider each siRNA knockdown as a combinatorial knockdown of multiple genes. Hence, every large-scale RNAi screen, though unintended, contains large

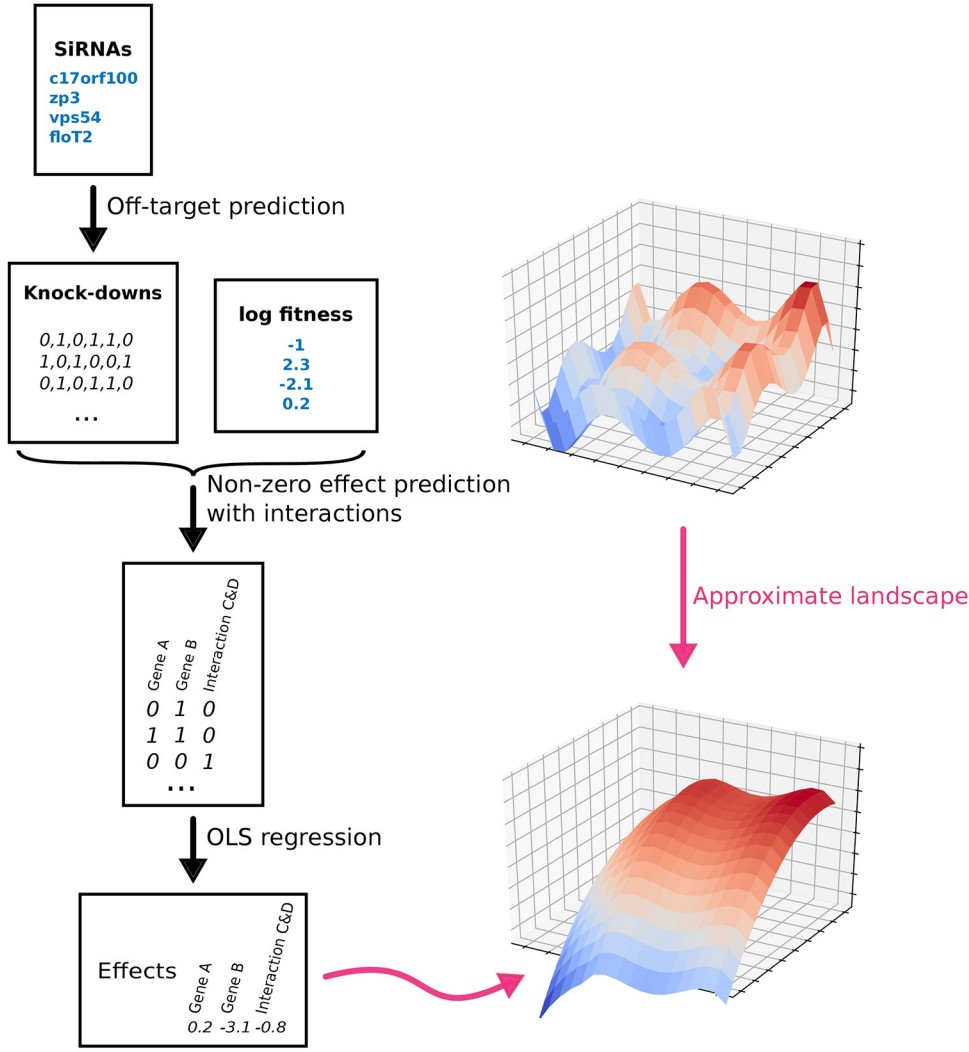

**Fig 1. RNAi fitness landscape model.** Black arrows indicate outputs that are actually produced. Red arrows indicate theoretical output.

numbers of observations of high-order combinatorial knockdowns and provides a rich source for the extraction of pairwise epistasis. These off-target effects have previously been used to improve inference of signalling pathways among a small number (on the order of a dozen) genes [22, 23]. Here, however, we attempt to use it to discover epistatic gene pairs in a genome-wide fashion (i.e. among tens of thousands of genes). Our approach is formulated as a regularized regression model. It can also be deployed for the estimation of epistasis from phenotypes other than fitness, such as for instance phenotypes that measure the activity of disease-relevant pathways, e.g. for pathogen entry [24], TGF$\beta$-signalling [25], or WNT-signalling [26]. Long term, the identification of disease-relevant epistatic gene pairs may allow the design or re-purposing of agents for combinatorial therapy with the potential to improve the efficacy of drugs.

In solving this model, we adapt two recent statistical learning methods, namely `glinternet` [27] and `xyz` [28] to select genes and gene-pairs with non-zero effects on fitness, and

evaluate both models on simulated data from real RNAi libraries. We vary the signal-to-noise ratios, number of true gene–gene interactions, number of observations per double knockdown and effect size for epistasis. We find that, within ranges that are realistic to real RNAi data, both approaches are capable of inferring pairwise epistasis with favourable precision and sensitivity when only a small number of genes are involved in interactions. In several tests `glinternet` continued to infer correct interactions up to several thousand genes, however the run time prohibits more thorough testing. To demonstrate the model on a real data set, we use the perturbation data from [24]. Using `glinternet`, we search for interactions between kinases, and report the most significant results.

Our simulations are performed using `R`, and the source code is available at: https://github.com/bioDS/xyz-simulation.

## 2 Methods

We fix the binary alphabet $\Sigma = \{0, 1\}$ representing the two possible states in a perturbation experiment. The value zero denotes the normal state of the gene (unperturbed wild type), whereas the value one indicates knockdown of the gene (perturbed). For $p$ genes we denote by $\Sigma^p$ the set of binary sequences of length $p$, indicating the perturbation status of each gene. Any subset $\mathcal{P} \subseteq \Sigma^p$ is called a perturbation space and its elements are called perturbation types. If the perturbations are genetic mutations, then the perturbation types are genotypes.

### 2.1 Fitness landscapes and epistasis

In the following, we focus on fitness landscapes, but would like to note that the theory also holds for any mapping of perturbation type to phenotype. A fitness landscape is a mapping $f : \mathcal{P} \to \mathbb{R}_+$ from perturbation type space to non-negative fitness values. Genetic interactions are a property of the underlying fitness landscape [29]. For $p = 2$ genes, the perturbation type space $\mathcal{P} = \{0, 1\}^2$ contains the wild-type 00, two single perturbations 01 and 10, and the double perturbation 11. The fitness landscape $f : \{0,1\}^2 \to \mathbb{R}_+$ can be written as

$$
\begin{aligned}
f(0,0) &= \beta_0 \\
f(1,0) &= \beta_0 + \beta_1 \\
f(0,1) &= \beta_0 + \beta_2 \\
f(1,1) &= \beta_0 + \beta_1 + \beta_2 + \beta_{1,2}
\end{aligned}
$$

for parameters $\beta_i \in \mathbb{R}$. $\beta_0$ is called the bias, $\beta_1$ and $\beta_2$ main effects, and $\beta_{1,2}$ the interaction. Epistasis is defined as

$$\varepsilon = f(0,0) + f(1,1) - f(0,1) - f(1,0) \tag{1}$$

It measures the deviation of the fitness of the double knockdown from the expectation under a linear fitness model in the main effects. We see that $\varepsilon = \beta_{1,2}$.

**2.1.1 Fitness landscape model.** It is challenging to generalize the notion of epistasis (Eq 1), because in higher dimensions, many more types of genetic interactions exist [29], even when restricting to pairwise interactions. In general, it will be impossible to estimate all interactions encoded in the fitness landscape reliably from data. In the following, we show how to assess marginal and conditional pairwise epistasis. For $p \geq 1$ genes, we consider the Taylor expansion of the fitness landscape

$$f(x_1, \ldots, x_p) = \beta_0 + \sum_i x_i \beta_i + \sum_{i<j} x_i x_j \beta_{i,j} + \sum_{i<j<k} x_i x_j x_k \beta_{i,j,k} + \ldots \tag{2}$$

Ignoring interactions of order 3 and higher we obtain the more computationally tractable approximation:

$$f(x_1, \ldots, x_p) \approx \beta_0 + \sum_i x_i \beta_i + \sum_{i<j} x_i x_j \beta_{i,j} \tag{3}$$

We show in Appendix A in S1 Appendix that in the fitness landscape model (Eq 3), which contains all main effects and pairwise interactions, but no interactions of higher order, the interaction terms $\beta_{i,j}$ alone determine conditional and marginal epistasis of the fitness landscape. Note that although we are discussing the Taylor approximation of the fitness function, the resulting pairwise epistasis definition is identical to that in [15].

## 2.2 Estimation of epistasis from RNAi perturbation screens

In *in vitro* RNAi experiments cells are perturbed by reagents, such as siRNA, shRNA, and dsRNA [30], each targeting a specific gene for knockdown. In recent years, it has been shown [20] that siRNAs exhibit strong sequence-dependent off-target effects, such that, in addition to the intended target gene, hundreds of other genes are knocked down. Thus, we can regard siRNA perturbation experiments as combinatorial knockdowns affecting multiple genes simultaneously. On the basis of the fitness landscape model (Eq 3), we propose a regression model for the estimation of epistasis from RNAi data. This inference is only feasible because of the unintended combinatorial nature of siRNA knockdowns.

**2.2.1 Perturbation type space.**   For an RNAi-based perturbation screen, the perturbation type space $\mathcal{P} = \{g_1, \ldots, g_n\}$ is represented as the $n \times p$ matrix $X$ that contains $g_i$ in row $i$. Based on the nucleotide sequences of the reagents, perturbations can be predicted by models for micro RNA (miRNA) target prediction [31]. We use $X_1, \ldots, X_p$ to denote the $p$ column vectors of $X$ for genes $1, \ldots, p$ and denote by $X_i \circ X_j$ the column vector consisting of the element-wise products of the entries of $X_i$ and $X_j$. As a measure of fitness, we use the vector $Y \in \mathbb{R}_+^n$, denoting the number of cells present after siRNA knockdown.

**2.2.2 Regression model.**   We aim to estimate the conditional epistasis $\beta_{i,j}$ between the $\binom{p}{2}$ pairs of genes $(i, j) \in \{1, \ldots, p\}^2$ from all combinatorial gene perturbations in the screen represented in the $n \times p$ matrix $X$, and the $n \times 1$ vector of fitness phenotypes $Y$. Based on (Eq 3) we regress phenotype Y on perturbations X,

$$\mathbb{E}[Y \mid X] = \beta_0 + \sum_i X_i \beta_i + \sum_{i<j} (X_i \circ X_j)\beta_{i,j} \tag{4}$$

The estimated $\beta_{i,j}$ are interpreted as the expected change in the response variable $Y$ per unit change in the predictor variable $(X_i \circ X_j)$ with all other predictors held fixed [32]. From Corollary 1 it follows that estimates for marginal epistasis $\varepsilon_{i,j}$ can be obtained by multiplication of $\beta_{i,j}$ with the constant $2^{p-2}$.

**2.2.3 Inference.**   We aim to infer the regression parameters $\boldsymbol{\beta} = (\beta_0, \boldsymbol{\beta}_{\{i:i>0\}}, \boldsymbol{\beta}_{\{i,j:i<j\}})$. Since it is infeasible to directly perform least squares linear regression on the matrix containing all $\binom{p}{2}$ interactions, we use a two-stage process. First, we use either the group lasso regularisation package `glinternet` [27], or the `xyz` interaction search algorithm [28] to select non-zero interactions. This variable selection step is the main computational challenge.

When using `glinternet`, we infer parameters $\boldsymbol{\beta} = (\beta_0, \boldsymbol{\beta}_{\{i:i>0\}}, \boldsymbol{\beta}_{\{i,j:i<j\}})$ by minimising the squared-error loss function

$$\mathcal{L}(\boldsymbol{Y}, \boldsymbol{X}; \beta) = \frac{1}{2} \left\| \boldsymbol{Y} - \left( \beta_0 + \sum_i \boldsymbol{X}_i \beta_i + \sum_{i<j} (\boldsymbol{X}_i \circ \boldsymbol{X}_j) \beta_{i,j} \right) \right\|_2^2 \tag{5}$$

under the *strong hierarchy* constraint

$$\beta_{i,j} \neq 0 \text{ if and only if both } \beta_i \neq 0 \text{ and } \beta_j \neq 0. \tag{6}$$

This constraint allows conditional epistasis between gene $i$ and $j$, i.e., $\beta_{i,j} \neq 0$, only if both single-gene effects $\beta_i$ and $\beta_j$ are present and constrains the search space. Lim and Hastie ([27]) show that this model can be formulated as a linear regression model with overlapped group lasso (OGL) penalty [33], where, in contrast to the group lasso [34], each predictor can be present in multiple groups.

To perform the variable selection, `xyz` searches for pairs $(i, j)$ that maximise $Y^T X_i X_j$. These are the interaction effects that account for the largest component of the response $Y$. While `xyz` can be used directly to find the largest interactions, we used `xyz_regression` to estimate all interactions. `xyz_regression` solves the following elastic-net problem [28]

$$\min_{(\beta_0, \beta) \in \mathbb{R}^{p+1}, \theta \in \mathbb{R}^{p(p-1)/2}} \left[ \frac{1}{2n} \sum_{i=1}^{N} (y_i - \beta_0 - x_i^T \beta - w_i^T \theta)^2 + \lambda (P_\alpha(\beta) + P_\alpha(\theta)) \right], \tag{7}$$

where

$$W \in \mathbb{R}^{n \times p(p-1)/2} = (X_1 \circ X_2, X_1 \circ X_3, \ldots, X_1 \circ X_p, X_2 \circ X_3, \ldots, X_{p-1} \circ X_p) \tag{8}$$

is the matrix of interactions, and $\theta \in \mathbb{R}^{p(p-1)/2}$ is the vector of regression coefficients for pairwise combinations of columns in $W$.

$$P_\alpha(\beta) = (1 - \alpha) \frac{1}{2} ||\beta||_{\ell_2}^2 + \alpha ||\beta||_{\ell_1} \tag{9}$$

is the elastic-net penalty.

The parameter $\alpha$ decides the compromise between the ridge-regression penalty ($\alpha = 0$) and the lasso penalty ($\alpha = 1$). We left the default value of $\alpha = 0.9$. The solution is found iteratively, with only a particular set of beta values are allowed to be non-zero at each iteration. In every iteration, the beta values that violate the Karush–Kuhn–Tucker conditions (Eq 10) are added to this set.

$$KKT \text{ Conditions}: \ X^T(Y - X\beta) = \lambda s, \ s_i \in \begin{cases} \{1\} & \text{if } \beta_i > 0 \\ \{-1\} & \text{if } \beta_i < 0 \\ [-1, 1] & \text{if } \beta_i = 0 \end{cases} \tag{10}$$

Rather than being computed directly, these beta values are found using the `xyz` algorithm (See Appendix C in S1 Appendix for details). We followed the recommendation in [28] and used $L = \sqrt{p}$ projections to find the strong interactions. Our own tests in Appendix D in S1 Appendix also suggest that further projections do not improve performance.

Second, once the non-zero effects have been estimated using either `glinternet` or `xyz`, we construct a matrix $X'$ with all elements of the set $\{X_i | X_i \neq \boldsymbol{0}\} \cup \{X_i \circ X_j | X_i \cdot X_j \neq 0\}$ as columns, in an arbitrary order. We then fit $Y \sim X'\beta$ using R's `lm` least squares linear regression

to calculate the coefficient estimates and corresponding p-values, the latter being whether the value significantly deviates from zero according to a t-test with $n - k - 1$ degrees of freedom, where $k$ is the number of effects predicted to be non-zero and including in the final regression step. We adjust the p-value to control the false discovery rate with the method of [35], and refer to this adjusted value as the q-value. Given this two-step procedure, we do not expect these values to be the same as if they were calculated using the complete interaction matrix and it should be noted that these may be biased estimates [44].

### 2.3 Software

The overlapped group lasso for strongly hierarchical interaction terms is implemented in the R-package `glinternet` 1.0.10 by Lim and Hastie [27] and available through the *Comprehensive R Archive Network* (CRAN) at https://cran.r-project.org/web/packages/glinternet/. The `xyz` algorithm is implemented in `xyz` 0.2 by Gian-Andrea Thanei [28] available at https://cran.r-project.org/web/packages/xyz/. The simulations are run using a version of this software that also contains a trivial bug fix, available at https://github.com/bioDS/xyz-simulation. For the data simulation, analysis and visualisation, we used the R-packages `Matrix` 1.2.6, `dplyr` 0.4.3, `tidyr` 0.4.1 and `ggplot2` 2.1.0. All simulations are performed using `R` 3.2.4.

### 2.4 Simulation of RNAi data

The data simulation followed a three-step procedure. First, we simulate the siRNA–gene perturbation matrix $X$ based on real siRNA libraries. Second, main effects $\beta_i$ and conditional epistasis between pairs of genes $\beta_{i,j}$ are sampled. Based on $X$ and $\beta$, we then sample fitness phenotypes $Y$ from our model (Eq 3) and add noise to match specific signal-to-noise ratios [36]

$$\text{SNR} = \frac{\text{Var}\left(\mathbb{E}[Y \mid X]\right)}{\text{Var}\left(Y - \mathbb{E}[Y \mid X]\right)}. \tag{11}$$

Details for each step including parameter ranges are as follows.

We simulate siRNA–gene perturbation matrices based on four commercially available genome–wide libraries for 20822 human genes from Qiagen with an overall size of 90000 siR-NAs. First, we predict sequence dependent off-targets using TargetScan [37] for each siRNA as described in [21]. We threshold all predictions to be 1 if larger than zero and 0 otherwise. Then, we sample $n = 1000$ siRNAs from $\{1, \ldots, 90000\}$ and $p = 100$ genes from $\{1, \ldots, 20822\}$ without replacement and construct the $n \times p$ binary matrix $X$. Hence, each row $X_{i\cdot}$ then contained the perturbation type $g_i = (x_{i,1}, \ldots, x_{i,p})$.

We simulate $q \in \{5, 20, 50, 100\}$ non-zero conditional epistasis terms $\beta_{i,j}$ between genes $i$ and $j$ from all observed combinatorial knockdowns, i.e. if the simulated screen contained siR-NAs that target both genes. This is a necessary condition for the identifiability of $\beta_{i,j}$ as otherwise, according to the model (Eq 4), $\beta_{i,j}$ will be multiplied by a zero vector $X_i \circ X_j = 0$. The effect size of the $\beta_{i,j}$ is sampled from N(0, 2). In order to maintain a strong hierarchy, we subsequently simulate for each interaction $\beta_{i,j}$ both main effects $\beta_i$ and $\beta_j$. Further, we add $r \in \{0, 20, 50, 100\}$ additional main effects. The effect sizes of the main effects are sampled from N(0, 1), so that the variance in the response fitness phenotypes are split in a ratio of 1:2 between main effects and interactions.

In order to model synthetic lethal pairs, interactions with effect strength of −1000 (on log scale) are added to the simulated data. Since lethal interactions may occur with little or no main effect present [10], we allow these pairs to violate the strong hierarchy and do not add

main effects. This is done both for biological plausibility, and to evaluate the performance of `xyz` and `glinternet` under less ideal circumstances. Since only `glinternet` assumes the strong hierarchy, this scenario might favour `xyz`.

Based on simulated perturbation matrices $X$, simulated main effects $\beta_i$ and interaction terms $\beta_{i,j}$, we sampled fitness values with $\beta_0 = 0$ according to the fitness landscape model (Eq 3)

$$Y \sim \mathrm{N}\Big(\sum_i X_i \beta_i + \sum_{i<j}(X_i \circ X_j)\beta_{i,j}, \; \sigma^2 I\Big),$$

where we chose $\sigma^2$ for fixed SNRs $s \in \{2, 5, 10\}$.

## 2.5 Evaluation criteria

We focus the evaluation on the estimated parameters of the model, specifically the conditional epistasis terms, $\hat{\boldsymbol{\beta}}_{\{i,j;i<j\}}$, rather than the model's performance in predicting the fitness phenotypes $Y$. Given the ground truth of true conditional epistasis between gene $i$ and $j$, $\boldsymbol{\beta}_{\{i,j;i<j\}}$, we assess the performance of the model to identify epistasis, i.e., estimated non-zero coefficients $\hat{\beta}_{i,j}$, by computing the number of true positives (TPs), false positives (FPs) and false negatives (FN). Here, TPs represent the number of gene pairs $(i, j)$ such that $\beta_{i,j} \neq 0$ and $\hat{\beta}_{i,j} \neq 0$, FPs the number of gene pairs $(i, j)$: $\beta_{i,j} = 0$ and $\hat{\beta}_{i,j} \neq 0$ and FNs the number of gene pairs $(i, j)$: $\beta_{i,j} \neq 0$ and $\hat{\beta}_{i,j} = 0$. The performance is then summarised using the following measures

$$\begin{aligned} \text{precision} &= \frac{\text{TP}}{\text{TP} + \text{FP}} \\ \text{recall} &= \frac{\text{TP}}{\text{TP} + \text{FN}} \\ \text{F1} &= 2\frac{\text{precision} \times \text{recall}}{\text{precision} + \text{recall}} \end{aligned}$$

Furthermore, we investigate whether estimates $\hat{\beta}_{i,j}$ have the same sign as the ground truth conditional epistasis and we quantify the deviation of the magnitude from the truth. Where applicable, we also evaluate the effect of selection of only those $\beta_{i,j}$ which significantly deviate from zero on the model's performance.

## 3 Results

First, we evaluate the proposed approach to estimating epistatic effects from off-target perturbations on simulated data. The approach depends on a model able to detect non-zero pairwise interactions (Fig 1). Here, we evaluate the approach using two such alternative models, `glinternet` and `xyz`.

We evaluate the ability of both `xyz` and `glinternet` to identify epistasis between pairs of genes from RNAi screens on simulated data with $p = 100$ genes and $n = 1000$ siRNAs. Only for `xyz`, we also test larger data sets, with $p = 1000$ and $n = 10000$. We use off-target information from real siRNAs and investigate the performance for varying signal-to-noise ratios, number of true interactions, number of observations per double knockdown, and effect sizes for epistasis.

We perform a separate set of tests where we specifically assess the performance of the two methods to identify synthetic lethal interactions, the strongest negative interactions. For this purpose, we simulate a separate data set that contains additional synthetic lethal pairs of genes.

In this test, we attempt to identify only lethal interactions using `xyz` and `glinternet`, given increasingly large numbers of genes.

## 3.1 Identification of epistasis under varying conditions

Both `xyz` and `glinternet` are tested on a series of small simulated data sets. For each combination of parameters $q \in \{5, 20, 50, 100\}$, $r \in \{0, 20, 50, 100\}$ and $s \in \{2, 5, 10\}$, controlling the number of true interactions, the number of additional main effects, and the SNRs of the fitness phenotypes, respectively, we sample 50 independent data sets. `xyz` is tested on a series of larger data sets, with parameters $q \in \{50, 200, 500, 1000\}$, $r \in \{0, 200, 500, 1000\}$ and $s \in \{2, 5, 10\}$. Only 10 independent data sets are sampled in these cases. Each data set consists of the perturbation matrix $X$, phenotypes $Y$, true conditional epistasis $\beta_{i,j}$ and main effects $\beta_i$.

The distribution of the number of observations for pairwise knockdowns of gene $i$ and $j$ is shown in Fig 13 in S1 Appendix for an exemplary perturbation matrix $X$. While only a few genes have many observations, 87% of gene pairs are simultaneously perturbed by at least one siRNA. Note that the distribution seen in Fig 13 in S1 Appendix is similar for both $p = 100$ and $p = 1,000$ genes. We also find that number of additional main effects has relatively little impact on detecting interactions (Appendix B in S1 Appendix), and this value is kept constant during our tests. We select only estimates $\hat{\beta}_{i,j}$ with a magnitude significantly different from zero (q-value $< 0.05$). This significantly improves precision, at a slight cost to recall, using both `glinternet` and `xyz` (Fig 2).

**3.1.1 Number of double knockdowns per gene pair.** We fixed the number of additional main effects to 20 and investigated performance with respect to the number of double knockdowns per epistatic gene pair, i.e. siRNAs that target both genes (Fig 3). The results are largely similar for both `xyz` and `glinternet`. As expected, for increasing numbers of observations, we observe an increase in precision and recall with a steeper increase of precision compared to recall and decreased performance for higher number of true interactions. The number of true epistatic gene pairs primarily affects recall, which decreases for higher numbers of true non-zero $\beta_{i,j}$. For gene pairs with more than 80 observations of the double knockdown,

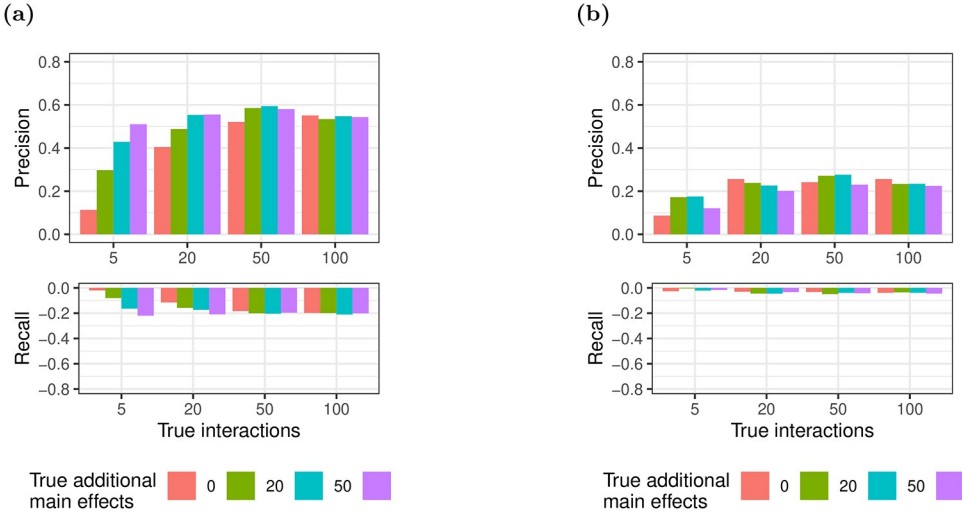

**Fig 2. Trade-off between precision and recall for selecting the subset of interactions significantly deviating from zero versus all interactions.** Top and bottom panels depict gain of precision and loss of recall, respectively. (a) `glinternet`; (b) `xyz`.

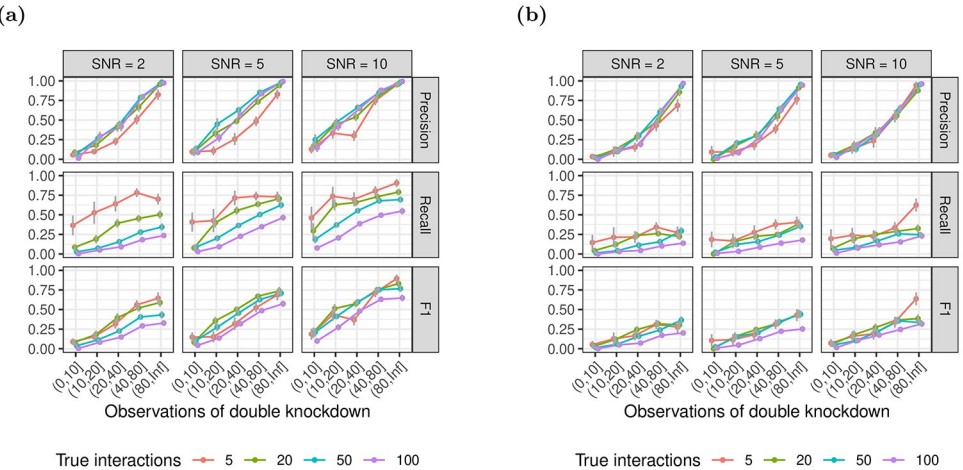

**Fig 3. Identification of epistasis for increasing numbers of observations of the pairwise double knockdown.** The number of additional main effects not overlapping with the set of interacting genes is fixed to 20. Results using (a) `glinternet` and (b) `xyz`.

`glinternet` shows strong performance with F1 values between $0.68 - 0.9$ across all tested numbers of true interactions and an SNR larger than or equal to 5 (Fig 3a).

`xyz` shows significantly improved performance for gene pairs with more than 40 observations, with F1 values almost all above 0.25. Small numbers of true interactions are particularly accurate, with $F1 > 0.5$ when there are also only 5 such effects (Fig 3b).

The number of times each pair of genes is observed is shown in Fig 4. We see that in the large simulation, in which all parameters are multiplied by ten, the number of observations of each pair of genes is similarly scaled. As a result, the overall distribution is similar to the smaller simulation.

**3.1.2 Epistatic effect size.** We observe that, for both `xyz` and `glinternet`, the performance of the model increases with the absolute value of the magnitude of the conditional

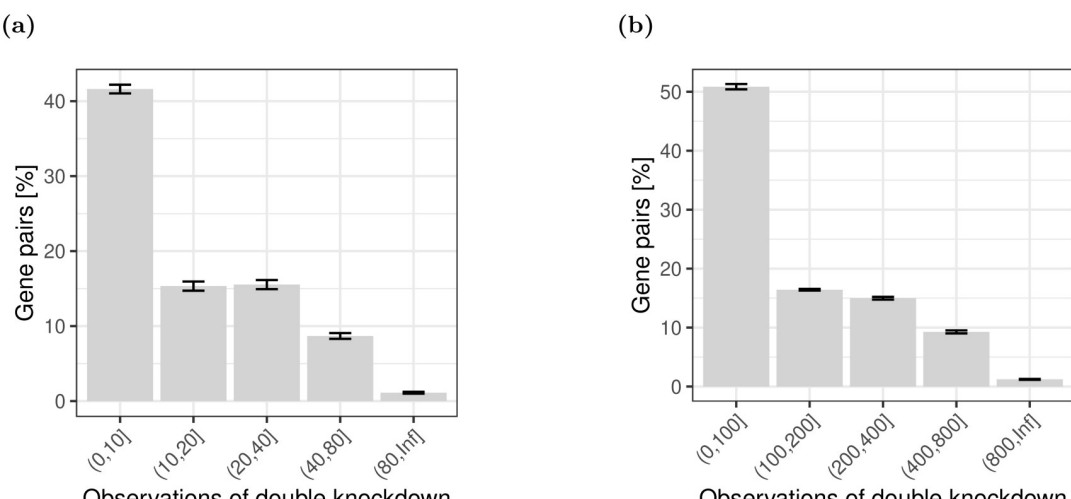

**Fig 4. The distribution of the fraction of gene pairs stratified by ranges of observed double knockdowns.** Gene pairs with zero observations are not shown. (a) $p = 100$, $n = 1000$; (b) $p = 1000$, $n = 10000$.

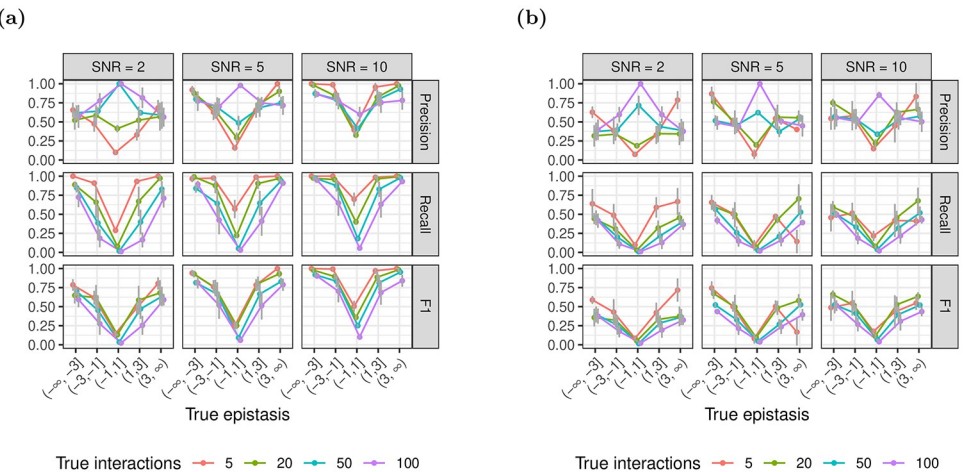

**Fig 5. Identification of epistasis for varying effect size.** Using (a) `glinternet` and (b) `xyz`.

epistasis between pairs of genes $|\beta_{i,j}|$ (Fig 5). Both for negative and positive epistasis, recall and precision steeply increase with increasing effect size. For pairs of genes with $|\beta_{i,j}| > 1$ and SNRs $\geq 10$, the model performs favourably with F1 values of 0.6 and higher in `glinternet`, and at least 0.25 in `xyz`. Overall performance also marginally improves for `glinternet` at SNR = 5, but no clear effect is seen for `xyz` or SNR = 10. With both `xyz` and `glinternet`, we observe exceptions to the general pattern of the overall V-shape for precision and recall, where strongly negative and positive epistasis and weak epistasis lead to high and low performance of the model, respectively. This effect can be explained by the fact that, after the significance test, an extremely small number of interactions are reported in these ranges (most often only one), with no false positives. The fact that the model's performance notably decreases for small effect sizes around zero explains why we observe a trend of decreasing performance for increasing numbers of true interactions, when we average over all effect sizes. This is because sampling true epistatic effect sizes from N(0, 2) for increasing numbers of true interactions increases the fraction of interactions with small effects around zero.

Notably, we can see in Fig 5b that even when the overall performance is poor, `xyz` is still able to find a small number of strong interactions relatively accurately. This is particularly promising, since synthetic lethal pairs would be such interactions.

**3.1.3 Direction.** We evaluate the ability of each method to distinguish between negative and positive epistasis among epistatic gene pairs identified as true positives (Fig 6). For both `glinternet` and `xyz`, the fraction of incorrect estimates of direction (positive vs. negative) is higher for decreasing effect size and increasing number of true interactions. For epistatic effects with an absolute value $> 1$, we observe at most 3% incorrect predictions with `glinternet`, and 8% with `xyz`. We observe at most 9% and 15% incorrect predictions for smaller effect sizes for `glinternet` and `xyz` respectively. Furthermore, we observe that increasing SNRs leads to a subtle decrease of incorrectly predicted direction.

**3.1.4 Magnitude.** We evaluate the deviation of the magnitude of estimates for epistasis from the ground truth as a function of observed double knockdowns (Fig 7). The deviation in magnitude is computed as $\frac{|\beta_{i,j}| - |\hat{\beta}_{i,j}|}{|\beta_{i,j}|}$, i.e. the percent relative change in deviation with respect to the true epistasis. We observe that across varying numbers of observations the model predicts the magnitude of epistasis between pairs of genes with high accuracy using both `xyz` and `glinternet`.

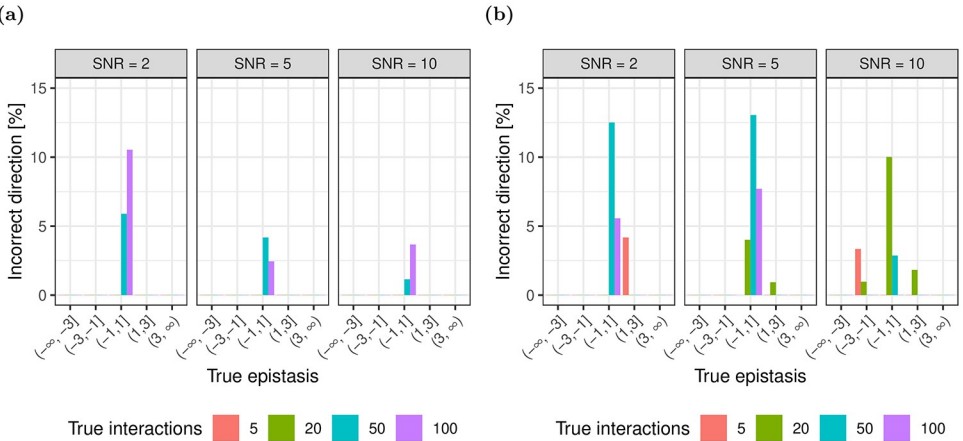

**Fig 6. Concordance between the sign of true and estimated epistasis.** The fraction of incorrectly identified signs between true and estimated epistasis for (a) `glinternet` and (b) `xyz`.

## 3.2 Scalability

Running `glinternet` until it has converged takes a prohibitively long time on larger data sets. While we are able to run our $p = 100$, $n = 1,000$ simulations in slightly under two minutes, increasing to $p = 1,000$, $n = 10,000$ takes over two days using ten cores. Although using more threads is possible, the running time is already dominated by single-threaded components with ten cores. The multi-threaded performance is therefore limited to by Ahmdahl's Law to approximately the performance we see here. Since fitting with small lambda values takes the majority of the time, we can improve this by changing the minimum value of lambda that gets used. Adjusting this from $\frac{\texttt{lambdaMax}}{100}$ to $\frac{\texttt{lambdaMax}}{4}$, and fitting only five lambdas in this range rather than fifty, `glinternet` still takes over an hour. This makes the repeated simulations from subsection 3.1 impractical at a larger scale with `glinternet`, although we do investigate some larger data sets in subsection 3.3.

It should also be noted that this limits the scale of real-world data that can be analysed using `glinternet`. While some improvements are possible by disabling cross-validation or

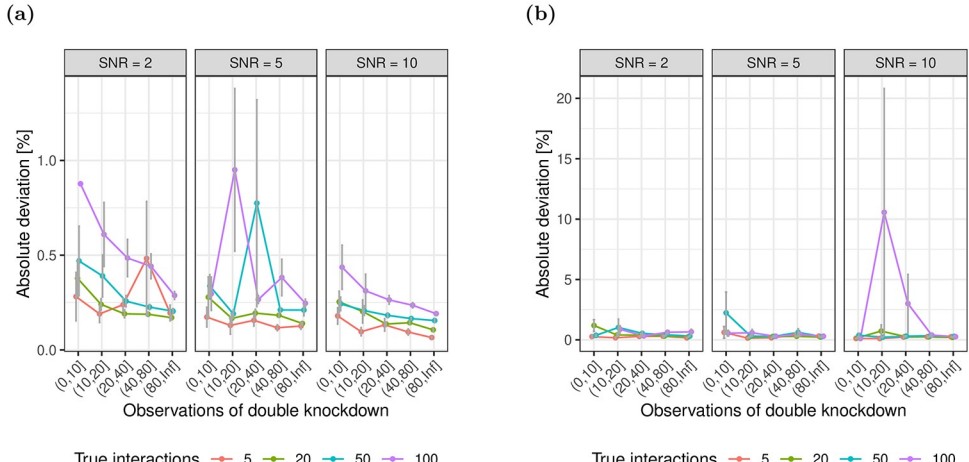

**Fig 7. Concordance between the sign of true and estimated epistasis.** The fraction of incorrectly identified signs between true and estimated epistasis for (a) `glinternet` and (b) `xyz`.

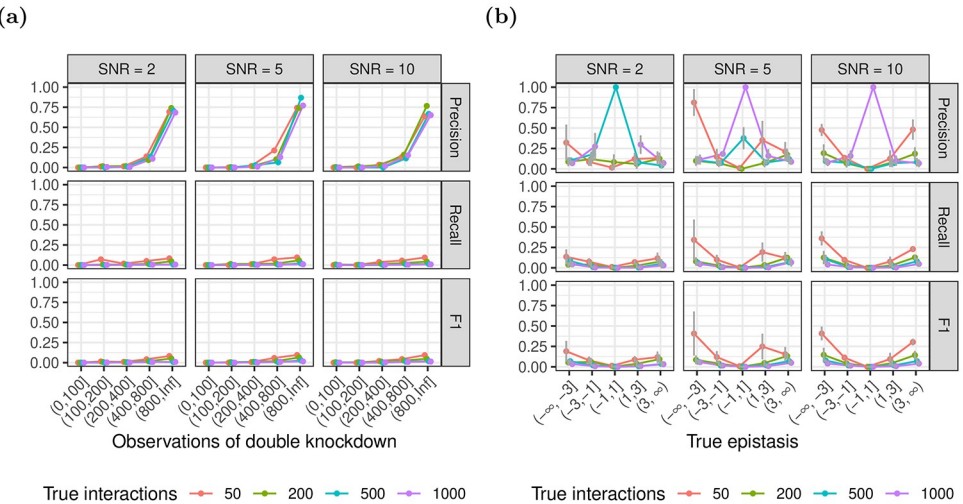

**Fig 8. Simulations repeated using `xyz` and larger data sets.** (a) number of observations of double knockdown. (b) Precision/recall/f1 by actual effect strength.

setting a high lower limit for lambda, we do not consider analysis of over 1, 000 genes to be practical. Further work needs to be done to develop regression methods for genome-scale data.

Since `xyz` has significantly shorter run time than `glinternet`, here we more thoroughly investigate performance on larger data sets. Repeating the earlier simulations with every parameter increased by a factor of 10 (Fig 8), we find that the overall trends remain the same. The fraction of incorrectly identified signs is omitted, as in this test there are no such results.

There is a significant drop in both precision and recall, and now only effects with a magnitude greater than 3 are found a significant amount of the time (Fig 8b).

## 3.3 Synthetic lethal pairs

Synthetic lethal pairs are of particular interest, and given that `xyz` is able to somewhat reliably find extremely strong interactions, it is natural to ask whether it can be used to quickly find lethal pairs, despite its poor performance on weaker interactions. We fix the number of main effects to 10, and simulated 10000 siRNAs on 1000 genes. Synthetic lethal pairs are created as interaction effects of magnitude −1000 (log scale). This rather extreme assumption makes synthetic lethals the best possible case for detection via regression. In practice, synthetic lethal detection accuracy would likely be somewhere between what we see here and that of a small negative effect. Since lethal pairs often do not have strong main effects (i.e. do not follow the strong hierarchy assumption), the components of the interaction are *not* used as main effects in this case.

Increasing the number of lethal interactions significantly reduces recall, but does not have a clear effect on precision. At this scale, `xyz` is often able to correctly identify some lethal interactions (Fig 9), particularly when there are only a few to find.

**3.3.1 Synthetic lethality detection in larger matrices.** While we could not run a significant number of tests at this scale using `glinternet`, we could investigate how well its accuracy scales compared to `xyz`. To do this, we simulated sets of up to $p = 4000$ genes, and measured the performance of both `xyz` and `glinternet`. In this case, both to avoid allocating more elements to a matrix than R allows, and to keep the run time of `glinternet` low, only $n = 2 \times p$ siRNAs are simulated. The ratio of siRNAs, genes, main effects, interactions,

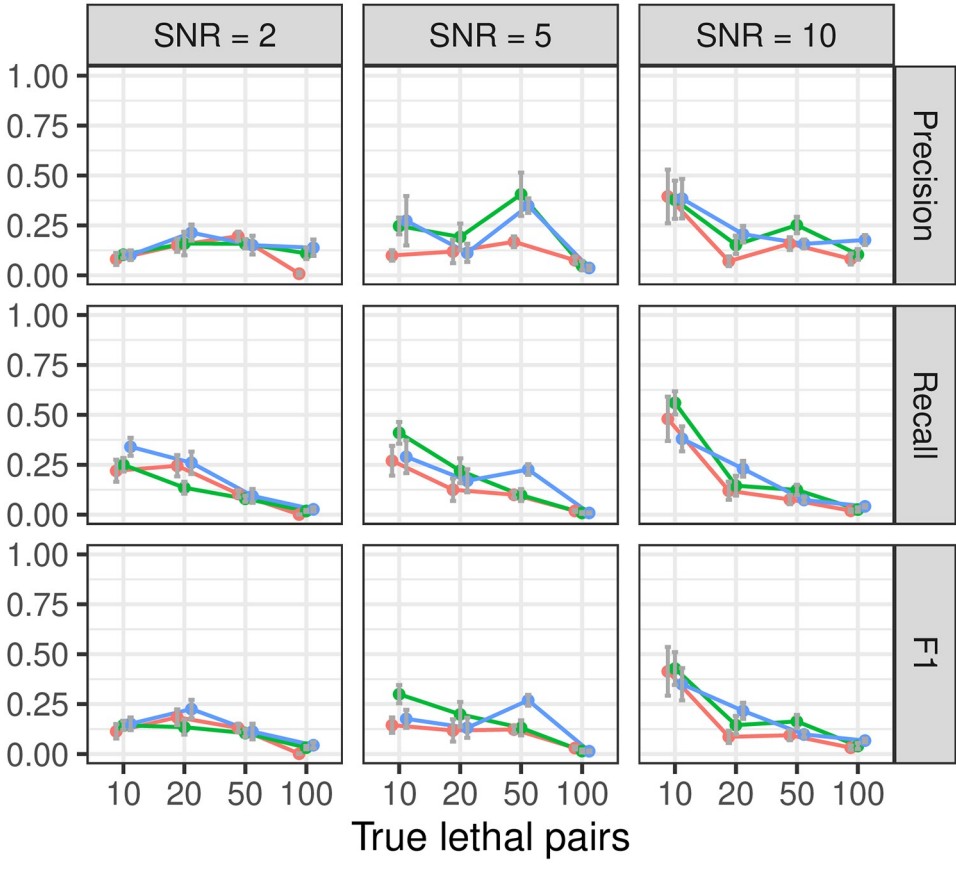

**Fig 9. Precision, recall, and F1 performance for varying numbers of synthetic lethal pairs, with additional background interactions, using `xyz`.** Neither side of the lethal interactions are used as main effects, and as far as lethal interactions are concerned, there is no hierarchy present.

and lethals, is fixed to: $n = 200$ siRNAs, $p = 100$ genes, $b_i = 1$ main effect, $b_{ij} = 20$ interaction effects, $l = 5$ lethal interactions. Data sets are then generated with these values multiplied by 5, 10, 20, and 40. As in the previous simulation, components of lethal interactions are *not* added as main effects. The strong hierarchy assumption is not valid in this case.

Interactions are then found with both `xyz` and `glinternet`. Here we focus specifically on synthetic lethal detection, and only correct lethal pairs are considered true positives, Any other pair (including a true interaction that is not a lethal) is considered a false positive.

We can see in Fig 10a that precision with `glinternet` remains fairly consistent as $p$ increases. There is a roughly proportional reduction in recall as the number of lethal interactions increases. After a slight increase from 500 to 100 genes, the actual number of significant interactions found remains fairly consistent. Beyond $p = 2000$, we found that `xyz` typically fails to find any of the lethal pairs (Fig 10b)

Fig 11 shows that neither `xyz` nor `glinternet` quite demonstrate a linear run time, but the run time for `glinternet` increases sharply beyond $p = 2000$. It is possible that this is simply the result of less efficient cache use with larger data, but it is nonetheless worth noting.

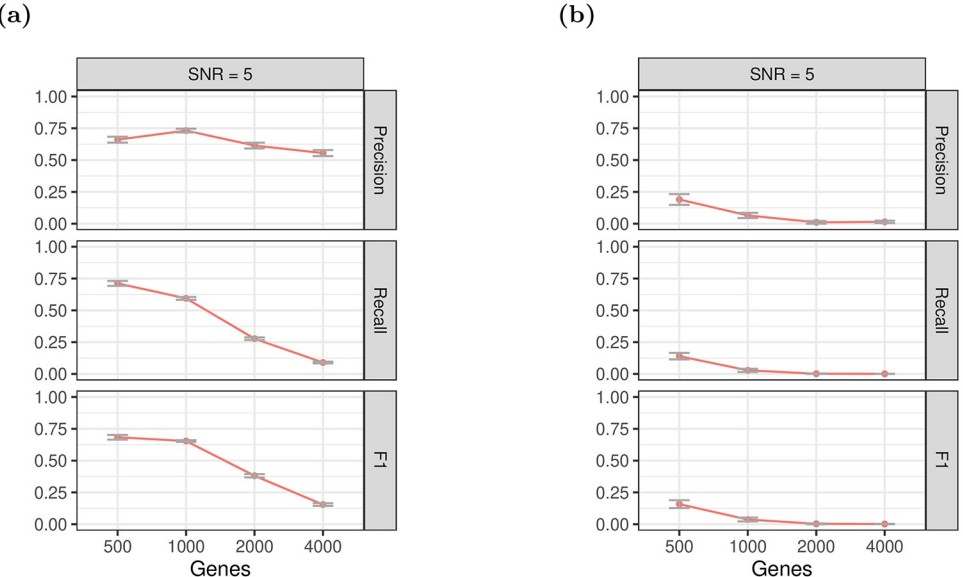

**Fig 10. Performance on increasingly large data sets.** (a) `glinternet` and (b) `xyz`.

## 3.4 Violations to model assumptions

For the regularised regression model (Eq 4) we assume strong hierarchy (Eq 6) between main effects $\beta_i$ and interaction terms $\beta_{i,j}$ in order to reduce the search space of all possible non-zero coefficients $p + \binom{p}{2}$ during inference. We refer the reader to [27], where Lim and Hastie show how violations to this assumption affect the performance. For instance, the performance of the model is evaluated when the ground truth only obeys weak hierarchy, i.e. only one main effect present, no hierarchy, or anti-hierarchy. Additionally, approximately 2.5% of simulations produced no interactions using `xyz`, because the estimated interaction frequency of non interacting pairs was too low. These were fairly evenly distributed across all combinations of parameters (Fig 12), and are not believed to have substantially affected the results.

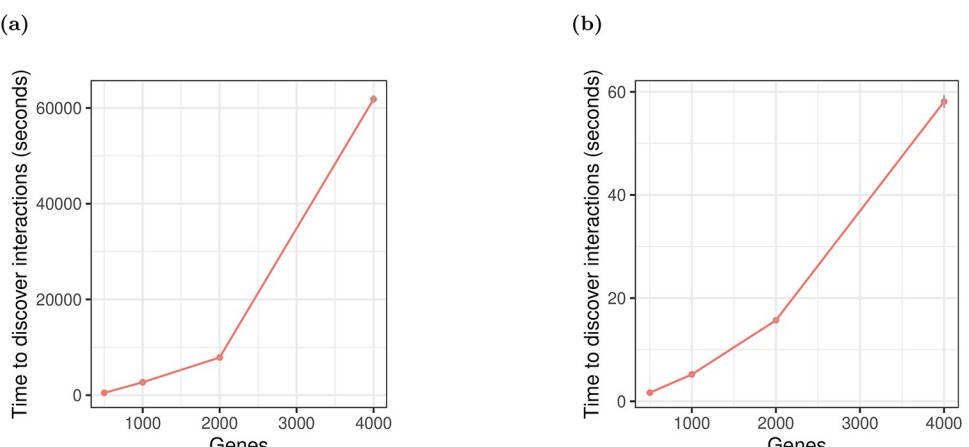

**Fig 11. Run time in seconds to find interactions on increasingly large data set.** (a) `glinternet`. (b) `xyz`. We compiled `glinternet` with OpenMP and ran with `numCores = 10`.

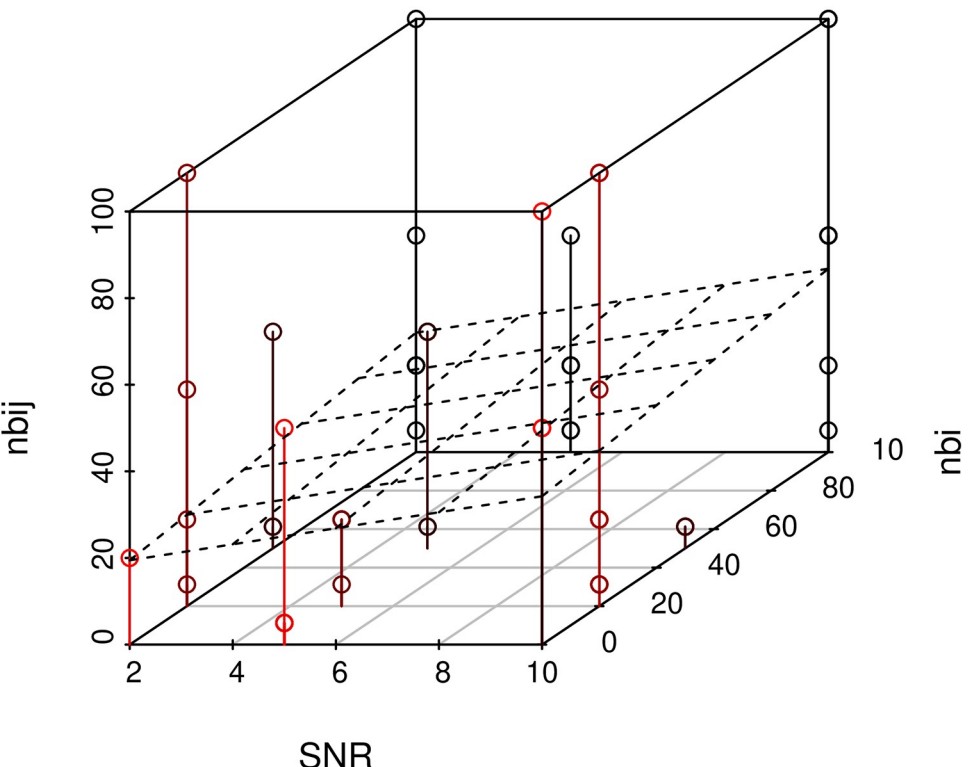

**Fig 12. Distribution of `xyz` failures.**

### 3.5 Summary recommendation

After simulating siRNA knockdown data sets of various sizes, and under various conditions, and attempting to reconstruct the interacting pairs using both `xyz` and `glinternet`, we arrive at the following recommendations. For data sets containing less than 4,000 genes (assuming between 2 and 10 experiments per gene), we recommend using `glinternet` to find interactions. Where `glinternet` would have a prohibitively long run time (data sets larger than those mentioned above), `xyz` continues to run quickly, and may still identify some useful results (Fig 9), particularly when interactions are observed a large number of times in the data and have strong effects (sections 3.1.1 and 3.1.2). Particularly when one expects a small number of significant interactions, increasing the number of projections beyond $\sqrt{n}$ may improve performance here (see Appendix D in S1 Appendix).

### 3.6 Effects in real data

Following the recommendation we have arrived at in subsection 3.5, we apply `glinternet` (followed by a linear regression analysis) to estimate epistatic effects from a real data set. We use the perturbation data from [24], containing siRNA screens targeting kinases in the presence of five bacterial pathogens and two viruses, and apply the routine as described in subsection 2.2 to identify pairwise kinase-kinase interactions. Specifically, we restrict the data to siRNAs that target kinases from the Qiagen Human Kinase siRNA Set V4.1, and the off-target effects within this set, resulting in an input matrix containing 11214 perturbations × 667 genes. Using $f = log_2\left(\frac{\text{Cells after}}{\text{Cells before}}\right)$ as a fitness measure, we found 1662 effects, 116 of which had a p-value less than 0.05. Since we have assumed that perturbations are binary in our simulations,

**Table 1. Ten most significant predicted effects of siRNA perturbation screens, targeting all human kinases.**

| Gene $i$ | Gene $j$ | Type | Combined Effect | P-value | $i$ effect | $j$ effect |
|---|---|---|---|---|---|---|
| CDK5R1 | RPS6KA2 | interaction | 12.52 | 0.0047 | 1.71 | -2.32 |
| RIPK4 | GRK3 | interaction | -3.24 | 0.0056 | -24.5 | 1.87 |
| PHKB | GUK1 | interaction | -7.47 | 0.0061 | 6.23 | -28.4 |
| MAP2K6 | UCK1 | interaction | -40.89 | 0.0094 | 13.8 | -22.6 |
| TNIK | PANK4 | interaction | -37.41 | 0.0115 | 21.3 | 5.21 |
| RPS6KB2 | TTK | interaction | 172.04 | 0.0118 | 0.5 | -20.4 |
| MAPK4 | TRPM7 | interaction | 9.49 | 0.0120 | 8.46 | 16.3 |
| HIPK1 | NUAK2 | interaction | -13.17 | 0.0126 | 18.1 | 29.5 |
| CDK19 | NA | main | 3.80 | 0.0136 | 3.80 | |
| C17orf75 | MAPK8IP3 | interaction | 21.74 | 0.0136 | 5.91 | 20.4 |

we continue to do so here. As a result, all non-zero predicted off-target effects are given a value of 1. The ten most significant predicted effects are shown in Table 1 (the full set of results, significant or otherwise, can be found at https://github.com/bioDS/xyz-simulation/blob/master/real_data_results_sorted.csv). Interestingly, the most significantly associated pair of genes, CDK5R1 and RPS6KA2, are both related to a common pathway. Specifically, CDK5R1 activates CDK5, which, along with RPS6KA2, is part of the IL-6 signalling pathway [38]. Searching both the ConsensusPathDB database [39], and STRING database [40] for relations between the found pairs, we find that a number of the interactions suggested here could be the result of existing known interactions. We each of the identified pairs of genes, we searched for common neighbours (a third gene with which both interact), shared pathways, and whether the produced proteins are found in the same protein complexes, and found the following known relationships:

CDK5R1 and RPS6KA2 share a common neighbour, and are present in four of the same enriched pathway-based sets. TTK and RPS6KA2 share nine common neighbours. RIPK4 and GRK3 share one neighbour, nd homologs were found interacting in other species. TNIK and PANK4 share one neighbour, as do MAPK4 and TRPM7, MAP2K6 and UCK1, and HIPK1 and NUAK2. As we could not locate the other identified pairs in the database, we hypothesise that they might constitute novel interactions.

Of the interactions present in Table 1, we see that HIPK1 and NUAK2, TNIK and PANK4, and MAP2K6 and UCK1 are predicted to have negative epistatic effects, and may be promising synthetic lethal candidates.

For comparison we also fit a linear model including all genes, but no interactions. Comparing the $R^2$ values for each, we find that individual gene effects explain $\approx$15% of the variance ($R^2 = 0.150$) Including the interactions chosen by `glinternet`, and removing the main effects it sets to zero, we have $R^2 = 0.392$, more than doubling the fraction of explained variance. The Adjusted $R^2$ is also significantly higher for the pairwise model, 0.286 as opposed to 0.096, indicating that the additional interaction variables are contributing significantly more than random. Moreover, the Akaike An Information Criterion (AIC) values indicate the pairwise model is more informative, with an AIC value of $-11607$ as opposed to the main effect only model's $-9853$. This highlights the importance of accounting for interactions in large-scale genotype-phenotype analyses, and relevance of bioinformatic tools with this capability.

## 4 Discussion

To the best of our knowledge, the presented model is the first approach that leverages the combinatorial nature of RNAi knockdown data resulting from sequence-dependent off-target

effects for the large-scale prediction of epistasis between pairs of genes. To do this, we take the second-order approximation of the fitness landscape, including only individual and pairwise effects, and attempt to infer the parameters of this model. Since `glinternet` is able to find pairwise interactions among $p = 1,000$ genes, we speculate that searching for three-way interactions is feasible among $\sqrt[3]{1,000^2} = 100$ genes. We are not aware of any software currently able to do this, however.

For the majority of our tests, we simulate the presence of a strong hierarchy. This constraint would imply that for the inference of non-zero epistatic effects between gene $i$ and $j$, $\beta_{i,j}$, we penalise cases where the main effects for both single genes, $\beta_i$ and $\beta_j$, are zero. This constraint significantly decreases the complexity of the search space of interactions. However, in biology there are many examples of epistasis where the marginal effects of individual genes are very small, for instance if both genes redundantly execute the same function within the cell [41]. [13] found in their study of experimental double knockouts in yeast that single mutants with decreasing fitness phenotypes tended to exhibit an increasing number of genetic interactions. This observation is reassuring for `glinternet`, which can pick up the interaction as long as the true single-mutant effects are not exactly zero. Moreover, Lim and Hastie showed in a simulation study that the model is in fact flexible enough to also identify pairwise interactions violating the strong hierarchy constraint [27]. For the detection of strong interactions, specifically synthetic lethal pairs, we have also demonstrated that the strong hierarchy constraint is not required.

In a simulation study, we sampled perturbations for $n = 1000$ siRNAs and $p = 100$ genes, and $n = 10000$ siRNAs with $p = 1000$ genes. As a consequence of high-throughput genome-wide screening platforms, the setting of $n = 10 \times p$, i.e. ten perturbations with different siRNAs per gene, is realistic even for higher order organisms with tens of thousands of genes [21, 24]. Sampling the perturbations directly from commercially available RNAi libraries allows us to translate results from the simulation study to applications on real data. We observe that increasing SNRs, as expected, results in an overall increase of the number of successfully identified gene pairs with true epistasis.

Nevertheless, we found that even for a moderate SNR of only 2, the model identifies interactions with acceptable performance using `glinternet` (F1 > 0.5 for 50 true interactions), when we observed each double knockdown over 40 times (Fig 3a) or the effect size of epistasis is larger than 1, i.e. $|\beta_{i,j}| > 1$ (Fig 5a). For an SNR of 5 and across all tested numbers of additional gene pairs and epistatic effect sizes, the performance of the model is approximately constant at around F1 = 0.5, independent of the number of true epistatic gene pairs (Appendix B in S1 Appendix).

Performance in our simulations also suggests that `xyz` is unable to accurately identify interactions in large data sets. Although `xyz` has a consistently short run time, and appears capable of running on genome-scale data, we see a significant drop in all other performance measures beyond $p = 1000$ genes.

The results when using `glinternet`, however, suggest that the general approach is able to accurately identify pairwise epistasis from large-scale RNAi data sets, given that the SNR of measured fitness phenotypes is larger than 2 and the effect size of epistasis is larger than 1. It is challenging to compare the performance of these models to approaches that estimate genetic interactions from other data, such as for instance from double knockout experiments [13], due to different scales of the epistatic effect size, however, the high precision of `glinternet` seems quite competitive. Moreover, our simulations demonstrated that if true epistatic effects between pairs of genes are identified, the model identifies both the direction of epistasis

(positive and negative) as well as the magnitude of the epistatic effect with high accuracy (Figs 6 and 7).

In detecting lethal interactions specifically, the high precision of `glinternet` after testing for significant deviations is particularly promising. Using this as a method to detect likely synthetic lethal interactions from RNAi data sets, we could propose candidates for further investigation as anticancer drug targets [9, 12]. While the run time may prevent `glinternet` from being used as such a method in genome-scale applications, we can recommend it for use with smaller data sets, or where the number of potential interactions can be significantly reduced prior to running `glinternet`. As the precision does not appear to suffer with larger input, only the run time, we believe combining linear regression with a perturbation matrix is a promising method for further investigation, and work to improve the performance sufficiently for use in genome-scale applications is ongoing.

Demonstrating our method on a set of kinase siRNA knockdowns, we find a number of plausible proposed effects (Table 1). This set is sufficiently small that the true positives may be found experimentally by testing all ≈1.4 million gene pairs (as in [13]). Alternatively, the most likely candidates may analysed with targeted sequencing and fitness measurements (as in [42]) or clinical trials, (as in [43]). It is likely that a significant number of false positives are present among the proposed interactions, and we consider such verification to be an essential second step in discovering true epistatic effects.

While filtering results by p-value does significantly increase accuracy (Fig 2), the p-values we use do not account for the prior variable selection (using `glinternet` or `xyz`), and may therefore be biased. Recent work is able to overcome this limitation with regard lasso regression in some cases [44, 45], however existing implementations [45, 46] of these methods require storing the full interaction matrix $X'$. For non-trivial numbers of interactions this does not typically fit in memory, and we cannot work with it directly. Moreover, the procedure from [47] is not applicable when $p \gg n$ unless the variance can be efficiently estimated. Given recent progress in variance estimation for lasso regression [48] it may be possible to implement unbiased p-value calculations for lasso regression at this scale, and we suggest this as one possible improvement for future work.

Finally, it is worth noting that this approach is not limited to siRNA perturbation matrices, or to synthetic lethal detection. Any method of suppressing gene expression, combined with an affected proxy for fitness, could be used to find likely candidates for epistasis.

## Supporting information

**S1 Appendix.**
(PDF)

## Author Contributions

**Conceptualization:** Ewa Szczurek, Niko Beerenwinkel, Alex Gavryushkin.

**Data curation:** Jeremy Jenkins.

**Formal analysis:** Kieran Elmes, Fabian Schmich, Niko Beerenwinkel, Alex Gavryushkin.

**Funding acquisition:** Niko Beerenwinkel, Alex Gavryushkin.

**Investigation:** Kieran Elmes, Fabian Schmich, Ewa Szczurek, Jeremy Jenkins, Niko Beerenwinkel, Alex Gavryushkin.

**Methodology:** Kieran Elmes, Fabian Schmich, Ewa Szczurek, Niko Beerenwinkel, Alex Gavryushkin.

**Project administration:** Niko Beerenwinkel, Alex Gavryushkin.

**Resources:** Jeremy Jenkins, Niko Beerenwinkel, Alex Gavryushkin.

**Software:** Kieran Elmes, Fabian Schmich.

**Supervision:** Ewa Szczurek, Niko Beerenwinkel, Alex Gavryushkin.

**Validation:** Kieran Elmes, Fabian Schmich.

**Visualization:** Kieran Elmes, Fabian Schmich.

**Writing – original draft:** Kieran Elmes, Fabian Schmich, Ewa Szczurek, Jeremy Jenkins, Niko Beerenwinkel, Alex Gavryushkin.

**Writing – review & editing:** Kieran Elmes, Fabian Schmich, Ewa Szczurek, Niko Beerenwinkel, Alex Gavryushkin.

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
