## [Decision Letter · Decision Letter 0]

7 May 2021

PONE-D-20-30514

Learning epistatic gene interactions from perturbation screens

PLOS ONE

Dear Dr. Gavryushkin,

Thank you for submitting your manuscript to PLOS ONE. After careful consideration, we feel that it has merit but does not fully meet PLOS ONE’s publication criteria as it currently stands. Therefore, we invite you to submit a revised version of the manuscript that addresses the points raised during the review process.

The reviewers found the method of interest but also raised a number of major limitations that require a detailed response by the authors. Importantly, reviewer 2 expressed concerns about the appropriateness of the statistical analysis.

We look forward to receiving your revised manuscript.

Kind regards,

Ruben Artero, Ph.D.

Academic Editor

PLOS ONE

Journal Requirements:

[This work has partially been funded by SystemsX.ch, the Swiss Initiative in Systems Biology,under IPhD grant 2009/025 and RTD grants 51RT-0126008 (InfectX) and 51RTP0151029(TargetInfectX), evaluated by the Swiss National Science Foundation.We acknowledge support from the Royal Society Te Ap ¯arangi through a Rutherford DiscoveryFellowship (RDF-UOO1702) awarded to AG. This work was partially supported by Ministryof Business, Innovation, and Employment of New Zealand through an Endeavour Smart Ideasgrant (UOOX1912) and a Data Science Programmes grant (UOAX1932).]

 [Funded: The required information is available in the Acknowledgements section of the manuscript.]

4. Thank you for stating the following in the Financial Disclosure section:

[Funded: The required information is available in the Acknowledgements section of the manuscript.].   

We note that one or more of the authors are employed by a commercial company: Novartis Institutes for BioMedical Research

5. Please remove your figures from within your manuscript file, leaving only the individual TIFF/EPS image files, uploaded separately.  These will be automatically included in the reviewers’ PDF.

6. Please ensure that you refer to Figure 14, 17, 18, 19, 20, 21 and 22 in your text as, if accepted, production will need this reference to link the reader to the figure.

Reviewers' comments:

Reviewer's Responses to Questions

**Comments to the Author**

1. Is the manuscript technically sound, and do the data support the conclusions?

Reviewer #1: Yes

Reviewer #2: Partly

2. Has the statistical analysis been performed appropriately and rigorously? 

Reviewer #1: Yes

Reviewer #2: No

3. Have the authors made all data underlying the findings in their manuscript fully available?

Reviewer #1: Yes

Reviewer #2: Yes

4. Is the manuscript presented in an intelligible fashion and written in standard English?

Reviewer #1: Yes

Reviewer #2: Yes

5. Review Comments to the Author

Reviewer #1: This paper describes a technique for inferring epistatic interactions from siRNA perturbation screens. The main ideas are (1) to treat the off-target effects of each siRNA together with the on-target effects as a complex perturbation of many genes and then (2) to extract epistatic interactions from a collection of the complex perturbations using a sparse regression framework. The paper sets up this framework, conducts an extensive simulation study where this approach is implemented using two different existing methods for sparse (elastic net) regression, and provides a reanalysis of an existing siRNA dataset.

The contribution here is sound, and indeed a rather creative idea, and is certainly suitable for PLoSOne with a few minor corrections.

- The definition of marginal epistasis could be improved by defining it in terms of averages rather than a sums. Then the fitness of a genotype in the marginal fitness landscape would simply be the expected fitness of a genotype given its allelic state at the specified subset of genes, and Corollary 1 simplifies to \\epsilon_{i,j}=\\beta_{i,j} instead of \\epsilon_{i,j}=2^{p-2} \\beta_{i,j}.

- The authors should also disambiguate between their concept of marginal epistasis and the (different) marginal epistasis concept studied in e.g. Crawford et al. 2017 PLoS Genetics.

- Sparse regression (Lasso, elastic map, and their Bayesian analogs) has previously been applied both in the fitness landscape and the quantitative genetics literature, and the reader should be aware of how the current contribution fits into this broader context. I am thinking of things like Cai et al. 2011 "Fast empirical Bayesian LASSO for multiple quantitative trait locus mapping" in the quantitative genetics literature and Otwinowski and Nemenman 2013 and Poelwijk et al. 2019 in the fitness landscape literature, but there is additional work in both areas (especially in quantitative genetics). One important feature of L1 regularized regression is that the results are sensitive to the choice of basis, and the Taylor basis used here is one difference with most other treatments of epistasis via regularized regression.

- The re-analysis of the siRNA screen should be clarified. The current text says that the kinases being targeted are kinases of the pathogens, but it appears that these are in fact the kinases of the human host. More generally, the reader needs more background on the experimental design of this study in order to understand the reanalysis. In Table 1, the reader also needs to see the corresponding single mutation effects in order to understand what kind of epistatic interactions these are (e.g. which of these are synthetic lethals), and some of the estimated effects seem awfully large (e.g. 172.04 on a log 2 scale).

Minor additional comments:

- More precise notation would be helpful in Eqn 6

- pg. 6 Karush-Kuhn-Tucker conditions

- pg. 7 \\theta needs to be defined

- pg. 17 What does failed to run mean?This paper describes a technique for inferring epistatic interactions from siRNA perturbation screens. The main ideas are (1) to treat the off-target effects of each siRNA together with the on-target effects as a complex perturbation of many genes and then (2) to extract epistatic interactions from a collection of the complex perturbations using a sparse regression framework. The paper sets up this framework, conducts an extensive simulation study where this approach is implemented using two different existing methods for sparse (elastic net) regression, and provides a reanalysis of an existing siRNA dataset.

The contribution here is sound, and indeed a rather creative idea, and is certainly suitable for PLoSOne with a few minor corrections.

- The definition of marginal epistasis could be improved by defining it in terms of averages rather than a sums. Then the fitness of a genotype in the marginal fitness landscape would simply be the expected fitness of a genotype given its allelic state at the specified subset of genes, and Corollary 1 simplifies to \\epsilon_{i,j}=\\beta_{i,j} instead of \\epsilon_{i,j}=2^{p-2} \\beta_{i,j}.

- The authors should also disambiguate between their concept of marginal epistasis and the (different) marginal epistasis concept studied in e.g. Crawford et al. 2017 PLoS Genetics.

- Sparse regression (Lasso, elastic map, and their Bayesian analogs) has previously been applied both in the fitness landscape and the quantitative genetics literature, and the reader should be aware of how the current contribution fits into this broader context. I am thinking of things like Cai et al. 2011 "Fast empirical Bayesian LASSO for multiple quantitative trait locus mapping" in the quantitative genetics literature and Otwinowski and Nemenman 2013 and Poelwijk et al. 2019 in the fitness landscape literature, but there is additional work in both areas (especially in quantitative genetics). One important feature of L1 regularized regression is that the results are sensitive to the choice of basis, and the Taylor basis used here is one difference with most other treatments of epistasis via regularized regression.

- The re-analysis of the siRNA screen should be clarified. The current text says that the kinases being targeted are kinases of the pathogens, but it appears that these are in fact the kinases of the human host. More generally, the reader needs more background on the experimental design of this study in order to understand the reanalysis. In Table 1, the reader also needs to see the corresponding single mutation effects in order to understand what kind of epistatic interactions these are (e.g. which of these are synthetic lethals), and some of the estimated effects seem awfully large (e.g. 172.04 on a log 2 scale).

Minor additional comments:

- More precise notation would be helpful in Eqn 6

- pg. 6 Karush-Kuhn-Tucker conditions

- pg. 7 \\theta needs to be defined

- pg. 17 What does failed to run mean?

Reviewer #2: I would like to thank the editor for the possibility of reviewing this article entitled “LEARNING EPISTATIC GENE INTERACTIONS FROM PERTURBATION SCREENS” by Elmes et al.

The topic presented is attractive as there is still an unmet need for the understanding of the epistatic gene interactions and its role in developing new therapeutic approach to achieve synthetic lethality.

The abstract summarizes the entire content of the article. The figures and tables help in summarizing its content.

However, this article presents several weaknesses. Here you can find a list of general comments and revisions:

• I have concerns about the two-stage approach to identify interactions. Using linear model to calculate estimates and p values does not take in account the selection process of LASSO. Refitting the regression model after variable selection may lead to biased estimates. However, some advances have been done in this area (Taylor & Tibshirani, 2016; D. Lee, J. et al, 2016).

• Importance of interactions is selected based on p value, however the criteria should be effect size along with p value. Consequently, please, review the following sentence from page 8 “We are nonetheless able (…)” I would suggest to remove it.

• I would like the authors to explain if the impact of the scalability could be a problem in real world data.

• As a general comment, strong emphasis is done to simulated data but there is a need to further investigate the effects of this methodology in real world data.

• The last paragraph of the results section compares R2 of two models, however it is well-known that R2 increases when including more covariates. It is recommended to compare models with a complexity penalization approach, i.e, AIC.

• I would like to know if the authors studied the role of epistatic gene interactions in any specific disease. The presence of some clinical example could improve the quality of the article making it more easily readable.

• As a minor comment, when referring to the Normal distribution in an equation it is conventionally assumed to write it as N(�, �2) instead of Norm(�, �2)

6. PLOS authors have the option to publish the peer review history of their article (what does this mean?). If published, this will include your full peer review and any attached files.

Reviewer #1: No

Reviewer #2: No

---

## [Author Response · Author response to Decision Letter 0]

8 Jun 2021

This is included in the cover letter PDF file.

---

## [Decision Letter · Decision Letter 1]

29 Jun 2021

Learning epistatic gene interactions from perturbation screens

PONE-D-20-30514R1

Dear Dr. Gavryushkin,

We’re pleased to inform you that your manuscript has been judged scientifically suitable for publication and will be formally accepted for publication once it meets all outstanding technical requirements.

Kind regards,

Ruben Artero, Ph.D.

Academic Editor

PLOS ONE

Additional Editor Comments (optional):

Reviewers' comments:

Reviewer's Responses to Questions

**Comments to the Author**

1. If the authors have adequately addressed your comments raised in a previous round of review and you feel that this manuscript is now acceptable for publication, you may indicate that here to bypass the “Comments to the Author” section, enter your conflict of interest statement in the “Confidential to Editor” section, and submit your "Accept" recommendation.

Reviewer #1: All comments have been addressed

Reviewer #2: All comments have been addressed

2. Is the manuscript technically sound, and do the data support the conclusions?

Reviewer #1: Yes

Reviewer #2: Yes

3. Has the statistical analysis been performed appropriately and rigorously? 

Reviewer #1: Yes

Reviewer #2: No

4. Have the authors made all data underlying the findings in their manuscript fully available?

Reviewer #1: Yes

Reviewer #2: Yes

5. Is the manuscript presented in an intelligible fashion and written in standard English?

Reviewer #1: Yes

Reviewer #2: Yes

6. Review Comments to the Author

Reviewer #1: (No Response)

Reviewer #2: (No Response)

7. PLOS authors have the option to publish the peer review history of their article (what does this mean?). If published, this will include your full peer review and any attached files.

Reviewer #1: No

Reviewer #2: No

---

## [Editor Report · Acceptance letter]

2 Jul 2021

PONE-D-20-30514R1 

Learning epistatic gene interactions from perturbation screens. 

Dear Dr. Gavryushkin:

I'm pleased to inform you that your manuscript has been deemed suitable for publication in PLOS ONE. Congratulations! Your manuscript is now with our production department. 

Kind regards, 

on behalf of

Dr. Ruben Artero 

Academic Editor

PLOS ONE